# Lympho-Hematopoietic Microenvironments and Fish Immune System

**DOI:** 10.3390/biology11050747

**Published:** 2022-05-13

**Authors:** Agustín G. Zapata

**Affiliations:** 1Department of Cell Biology, Faculty of Biology, Complutense University of Madrid, 28040 Madrid, Spain; zapata@ucm.es; Tel.: +34-913-944-979; 2Health Research Institute, Hospital 12 de Octubre (imas12), 28041 Madrid, Spain

**Keywords:** teleost fish, immunity, thymus, peripheral lymphoid organs, MALT, T and B lymphocytes, germinal centers and immune memory

## Abstract

**Simple Summary:**

Teleost fish, the most abundant group of vertebrates, represent an excellent tool to establish possible correlations between the histological organization of their lymphoid organs and their immunological capacities. This approach allows us to analyze embryonic and larval lymphopoiesis, the remarkable organization of the teleost thymus, the role of the kidney as a true equivalent of the lympho-hematopoietic bone marrow of higher vertebrates, the mechanisms of antigen trapping in both ellipsoids and the so-called melano-macrophage centers (MMCs) and their relation with the generation of memory and the lack of germinal centers, and the extended development of the lymphoid tissue associated to mucosae.

**Abstract:**

In the last 50 years information on the fish immune system has increased importantly, particularly that on species of marked commercial interest (i.e., salmonids, cods, catfish, sea breams), that occupy a key position in the vertebrate phylogenetical tree (i.e., Agnatha, Chondrichtyes, lungfish) or represent consolidated experimental models, such as zebrafish or medaka. However, most obtained information was based on genetic sequence analysis with little or no information on the cellular basis of the immune responses. Although jawed fish contain a thymus and lympho-hematopoietic organs equivalents to mammalian bone marrow, few studies have accounted for the presumptive relationships between the organization of these cell microenvironments and the known immune capabilities of the fish immune system. In the current review, we analyze this topic providing information on: (1) The origins of T and B lymphopoiesis in Agnatha and jawed fish; (2) the remarkable organization of the thymus of teleost fish; (3) the occurrence of numerous, apparently unrelated organs housing lympho-hematopoietic progenitors and, presumably, B lymphopoiesis; (4) the existence of fish immunological memory in the absence of germinal centers.

## 1. Introduction

Almost 20 years ago, Louis du Pasquier concluded: “the only feature that underwent major changes during vertebrate evolution was the organization of lymphoid organs resulting in a better exploitation of the somatic events generating lymphocyte diversity in warm-blooded vertebrates” [1]. In the same way, more recently, Thomas Boehm [2] pointed out that the knowledge of the structure of primary and secondary lymphoid organs throughout the vertebrate phylogeny was critical to understand the sequential innovations that affect vertebrate adaptive immunity. In fact, the appearance of the adaptive immune system in jawed vertebrates is apparently the successful culmination of a process developed in the extinct Placoderm fishes but really is the beginning of a complex evolutionary process in which teleost lymphoid organs played a remarkable role.

It is important to remember that whole genome duplication is a major characteristic of the evolution of vertebrates. A first whole genome duplication gave rise to the ancestor of vertebrates, a second one resulted in the emergence of jawed vertebrates and, equally, a third genome duplication occurred at the origin of ray-finned fish [3]. Thus, in the ancestors of modern teleost fishes the number of loci related with the immune functions doubled [4], and, although only a small part of the duplicated genes was retained, the signature is still present in the genomes of modern teleosts [5].

On the other hand, there are almost 32,000 bony fishes. This group, originated 400 million years ago, includes around 31,000 species of modern teleosts, making it impossible to determine a common pattern on their structural and functional immunological capacities; but, it is possible to establish some common rules, particularly between some teleost species, such as zebrafish, medaka, tilapia, salmonids, etc., that have been greatly studied, in which we will examine specifically:

The organs containing lymphoid progenitors and their evolution along development (Section 2).

The thymocyte differentiation in an adult thymus that neither separates from the pharyngeal epithelium nor frequently exhibits a clear cortex-medulla demarcation (Section 3.1, Section 3.2, Section 3.3 and Section 3.4).

The B cell development in vertebrates that neither contain a hematopoietic bone marrow nor exhibit germinal centres in peripheral lymphoid organs (Section 3.5 and Section 3.6).

A peripheral lymphoid system devoid of lymph nodes and where mucosal immunity appears to be very relevant as a first defence line against aquatic pathogens (Section 3.7).

## 2. The Lympho-Hematopoietic Sites of Embryonic Teleost Fish and the Appearance of Their Primary and Secondary Lymphoid Organs

Blood precursor cells appear in different hematopoietic sites in embryonic and larval teleosts (Figure 1). Primitive embryonic blood cells originate in some teleost species in the extraembryonic yolk sac, whereas in other species, including anglerfish, killifish and zebrafish, the first hematopoietic site is the named intercellular mass (ICM), an intraembryonic dorsolateral area [6,7]. In this primitive first wave, the medial and anterolateral mesoderm produces embryonic erythroid and myeloid cells, respectively [8]. There is also a transient, intermediate wave in which erythroid–myeloid cell progenitors appear in the posterior blood islands (PBIs) [9]. Finally, definitive hematopoietic stem cells (HSCs) are formed for the first time in the ventral wall of the dorsal aorta (VDA), an equivalent of the aorta-gonad-mesonephros (AGM) region of higher vertebrate embryos, by a process of endothelial–hematopoietic transition [9,10,11,12]. HSCs from the VDA then migrate to the caudal hematopoietic tissue (CHT) from where they move to both the thymus and kidney, the main hematopoietic organs of adult teleosts. In addition to the generation of HSCs, it has been reported [9,12] that the hemogenic endothelia of both aorta and PBIs transiently produce HSC-independent T cells, and myeloid and erythroid cells, but not B cells.

The CHT was firstly described as a hematopoietic microenvironment largely consisting of the caudal aorta and the caudal vein plexus that, together with the perivascular mesenchymal stromal cells, form an hematopoietic niche that transiently accommodates HSCs migrated from the AGM and modulates their proliferation and differentiation [13,14,15,16]. In the CHT, endothelial cells express Kit-Lb, thrombopoietin, oncotastin M and CSF-3a that support HSC expansion and differentiation [16,17,18]. The CXCL18b/CXCR1 pair increases the time of homing and expansion of the hematopoietic stem progenitor cells (HSPC) in the CHT [19], whereas CCL25b/CCR7 interactions contribute to their migration from the CHT to the thymus after the activation of specific Rho GTPases, such as Rac2, involved in the F-actin cytoskeleton dynamics [20].

Accordingly, there are two successive waves of T lymphopoiesis in zebrafish: an early HSC-independent T lymphopoiesis that largely produces CD4^+^ TcRαβ cells, some of them expressing Foxp3α, in the larvae, and a late HSPC-dependent T cell production able to generate different T cell subsets in larval and adult zebrafish. Thus, HSC-independent T cells emerge in zebrafish as early as 3 days post-fertilization (dpf) remaining until the juvenile stage, whereas HSC-derived T cells appear later, around 8 dpf until adulthood [12]. In this same study, authors determined that VDA was also the principal source of γδ T cells [12]. What is the biological function of this early CD4^+^ T cell subset? As observed in murine fetal thymus [21], these cells could assume T regulatory (T reg) cell functions promoting immune tolerance [22]. However, this cell subset could perhaps be involved in the maturation of medullary TECs as the named CD4^+^ inductor lymphoid cells of murine embryonic thymuses [23].

In zebrafish, around 4 dpf, HSC begin to colonize the adult lympho-hematopoietic organs and at 5 dpf appear in pronephros (head/anterior kidney) and between the renal tubules of mesonephros (trunk/caudal kidney). At this hematopoietic site, HSCs renew, expand and differentiate to all blood cell lineages, including lymphoid progenitors, for the lifetime of the fish [7]. Danilova and Steiner [24] observed Rag-1 expression by ISH (in situ hybridization) in the pancreas of 4-day-old zebrafish, and Lam and colleagues [25], by RT-PCR in 3-day-old zebrafish, related this detection to the presence of immature B lymphocytes. However, these results were strongly questioned and, currently, B lymphopoiesis is assumed to occur in the teleost kidney, presumably in its anterior part [26].

The kidney organizes a mesenchymal stroma equivalent to the microenvironment of mammalian bone marrow [27,28] that supports the maintenance, expansion and differentiation of hematopoietic cells. In this regard, the teleost kidney exhibits the dual condition of the bone marrow acting as a hematopoietic site that homes HSPCs and a locus for B lymphopoiesis including immature but also mature effector and memory B lymphocytes [29]. Moreover, it has been suggested that CCL20-L1a, a chemokine highly expressed in the rainbow trout caudal kidney, would create a chemotactic gradient among the head and the trunk kidney for guiding B lymphocytes to leave the pronephros after reaching maturation [30].

In the adult condition, lymphoid progenitor cells leave the kidney and home into the thymus and later in the splenic primordium [2]. In some marine teleosts, the spleen has been reported to be organized earlier than the thymus [26]. In fact, splenic primordium in these species is really an erythropoietic organ rather than lymphoid, organizing its lymphoid architecture later when the antigen trapping structures are formed [6]. 

## 3. The Teleost Adult Lymphoid Organs

### 3.1. The Thymus: Histological Organization and Non-Lymphoid Cell Components

In the early somite stage, a group of NKx2.3^+^ cells under control of BMP2 signaling, determines the region of pharyngeal endoderm that, with the concourse of neural crest-derived mesenchyme cells from the third and fifth hindbrain rhombomeres [31], will constitute the zebrafish thymic primordium [32]. The thymus is the first organ to become lymphoid after receiving lymphoid progenitor cells from the AGM and/or the CHT, which, in the thymus, initiate the T cell differentiation following Rag-1 expression [33].

Adult teleosts are the unique vertebrates in which the adult thymus remains intimately associated to the original pharyngeal epithelium. In addition, in many teleost fish, it is difficult to distinguish a clear cortico-medullary demarcation, a typical signature of the thymus of higher vertebrates [34,35].

Thymic epithelium is closely associated with the pharyngeal cavity [36,37], where major histocompatibility complex (MHC) class II^+^ cells have been reported in sea bass [38] and rainbow trout [39]. This particular histological organization suggested to Tatner and Manning [40] that antigens present in the water could access directly into the embryonic thymus from the gill cavity. However, trout fries immersed in a ferritin solution neither trap ferritin through the gills nor through the thymic primordium but via the basement membrane lining the connective tissue capsule of the organ [41]. On the other hand, in several teleost fish, including zebrafish, turbot, sea bass, halibut, etc., a cortical area has been identified in the outer zone of the organ, whereas the medulla appears in the inner zone. However, in other fish, such as the flounder, rainbow trout and Atlantic salmon, a clear distinction is lacking [35].

It has been suggested that both aspects, the continuity of thymic tissue with the pharyngeal epithelium and the lack of a clear cortico-medullary separation reflect an incomplete migration of the thymic primordium from the gill buds. Thus, fish exhibiting a more internalized thymus could represent a first stage of the internalization that the organ undergoes during evolution [42]. More recently, we proposed that murine thymus develops, as other organs derived from the anterior endoderm (i.e., salivary glands, lungs, pancreas), according to a process of tubulogenesis and branching morphogenesis [43]. This process could be partially abolished in teleost, avoiding the separation of the thymic primordium from the branchial endoderm and the complete organization of the thymic medulla. Nevertheless, it is important to remark that in some fish, thymus shape is irregular with important differences along the dorsal–ventral and anterior–posterior axes that make it difficult to understand the histological sections [44,45]. Thus, frequently the cortex under both the connective tissue capsule and the pharyngeal epithelium surrounds the central medullary area totally, but in other sections, the area of densely packed thymocytes only occurs under the capsule (Figure 2) [35].

We identified four thymic areas in the trout thymus [46]: a subcapsulary area under the basal connective tissue capsule that contains lymphoblasts and appears to be equivalent to the subcapsulary area of the mammalian thymic cortex; an inner zone similar to the cortex and the outer zone, which largely contains layers of thymic epithelial cells (TECs), some of them in continuity with the pharyngeal epithelium, presumably similar to the medulla. Note that we consider the inner zone to be the most separated from the gill cavity and the outer zone to be close to the pharyngeal epithelium. However, other studies assume that the medulla occupies a more central position surrounded by a cortical outer zone. In fact, many morphological studies identify the cortex as the strongly stained thymic area that contains more thymocytes and the medulla a less stained region where the epithelium predominates. Other studies have detected molecules known to be specifically expressed in the cortex or medulla, to mark the location of these areas in teleost fish. Thus, in situ hybridization (ISH) was used to detect Rag-1 and CD3ɛ in the cortical-like, outer zone [47]. T-cell receptor (TcR) α transcripts appear scattered among TECs, weakly expressed in the outer cortex [45], where the cortical (c) TECs express Dll4, a Notch ligand, as in the murine thymus [48]. On the other hand, in the medaka thymus, the dorsal part corresponding to the medulla, where the thymocytes are less densely packed, express the UEA-1 lectin [49] and Aire, two medullary (m) TEC markers in the mammalian thymus [48]. 

A more important issue is to correlate these histologically characterized thymic compartments with their presumptive functions. Direct or indirect approaches allow, in general, to conclude that the teleost thymus contains the majority of the cellular components described in the organ of higher vertebrates and presumably functions similarly. As in mammals, thymic epithelium constitutes the major non-lymphoid component of the teleost thymus, although the lack of specific reagents does not allow a precise identification of the different cell types found in higher vertebrates. Ultrastructurally, only one type of TEC was reported in the thymus of *Rutilus rutilus* [50] and *Stcyases saguineus* [51], but seven TEC subsets were identified histochemically in rainbow trout [46]. 

Other thymic structures whose presence in teleosts is a matter of discussion are the so-called Hassall bodies or corpuscles. Whereas some authors have reported their existence [44,51,52,53], others have not found such structures [42,46,53,54]. Hassall bodies are epithelial multilayers morphologically similar to the epidermal layers of skin that occur in the thymus of higher vertebrates, particularly humans [55], but also have been related with the final differentiation of Aire^+^ mTECs and the presence of tuft cells [56,57]. We know little on these cell types in teleosts as only in a couple of studies have the presence of Aire^+^ cells and/or Aire transcripts been reported [48]. In any case, the number and development of these thymic epithelial structures are very variable even in the mammalian thymus and, in teleosts, they are presumably epithelial cysts in which the 3D organization of the medullary thymic epithelial network has disappeared [58].

Macrophages, dendritic cells (DCs), myoid cells and secretory, presumably neuroendocrine, cells also appear in the thymus of various teleost fish. Myoid cells are large, isolated striated muscle cells whose myofilaments organize “sarcomeres” around the nucleus. They are abundant in amphibians and reptiles but scarce in teleost fish [59]. Studies based on quail-chick chimaeras indicated that myoid cells derive from neural crest cells [60]. Their functional significance is unknown. Finally, numerous vacuolated, secretory cells were ultrastructurally described in the teleost thymus [54,61]. DCs have also been observed in the teleost thymus [48].

### 3.2. The Thymus: The Lymphoid Cell Components 

T cell maturation has been extensively analyzed only in a few teleost species, particularly zebrafish, medaka, cyprinids and rainbow trout, due to the lack of specific reagents, and to extrapolate the results needs caution.

The relevance of the thymus for T cell differentiation was robustly supported, despite the technical difficulties, by seminal studies on early thymectomized fishes, in which it was possible to demonstrate important deficits in the T cell responses [62,63]. 

Teleost T cells express, with subtle differences, the most molecules routinely used in mammals to define, at least, the major T cell; in addition, we know their responses to different agents, but it is more difficult to establish direct connections between expressed phenotypes and specific functions. Fish T cells express somatically rearranged T-cell receptor (TcR) α and β chain genes as well as TcRγ and δ genes [64,65]; they contain all CD3 components, express CD4 and CD8 co-receptors and co-stimulatory molecules such as CD28, CTLA-4 and CD80/CD86, but presumably neither ICOS nor PD-1 [64,65].

In teleosts, there are two CD4 molecules (CD4-1, CD4-2) [66] that, together with MHC class II genes, are highly expressed in the thymus [38]. Peripheral CD4-1^+^ cells express the transcription factor T-bet and IFNγ and could represent TH1-like cells; others express IL17A/F supporting the existence of teleost TH17-like cells [67] and finally other CD4-1^+^ cells express the regulatory molecules Foxp3, TGFβ and IL10 [68]. In the case of CD4-2^+^ cells, some cells express IL15Rα, whereas others express Foxp3 [69]. The presence of TH2-like cells in teleost fishes has been related with the expression of Gata3, IL4/IL13A and B in CD4-1^+^ cells [70]. As expected, only fish thymocytes co-express CD4 and CD8 [71,72].

γδ T cells of teleost thymuses have similar characteristics to those of mammals. γδ T cells in medaka occupy the subcapsulary region of the thymic cortex and the central medulla. Lymphoid progenitors arriving to the medaka thymus progress to the γδ T cell subsets; out from this thymic niche favoring γδ T cell development, the progenitor cells differentiate to αβ T cells with the concourse of IL7 and CCL25a (see below) [73]. In addition, it has been indirectly suggested that ectothermic vertebrates do not have a pTcRα gene necessary in higher vertebrates for the maturation of double negative (DN) thymocytes to the double positive (DP) cell compartment [74].

### 3.3. The Thymus: Mechanisms That Govern Both Phenotypical T Cell Differentiation and Thymocyte Education

Molecules regulating the behavior of lymphoid progenitors toward and inside the thymus, as in mammals, include chemokines (QKs) and their receptors (QKRs), cytokines, largely IL7, and the Notch/Notch L signaling pathway, under the control of the family of the transcription factors Foxn that regulates the essential development of TECs (see below). Firstly, fish thymopoiesis is regulated by the chemokine-dependent migration of lymphoid progenitor cells from the AGM region and CHT into the thymic primordium [33]. CCL25 and CXCL12 attract these progenitors in the majority of teleosts [75]. The blockade of CCL25 signaling, the unique QK in the medaka thymus [75], impedes the homing of lymphoid progenitors into the thymic rudiment [76]. Accordingly, any factor, such as Arf4a (Auxin response factor 4a) or Ikzf1 (Ikaros family Zn finger 1), that modulates the activation of CCR9, the CCL25 receptor, regulates the migration of lymphoid progenitors into the teleost thymus [77]. CCR9a and Arf4a are critical factors for early T cell lymphopoiesis and direct targets for the action of Ikzf1, a transcription factor associated with chromatin remodeling. Remarkably, CCR9 and/or Arf4a expression in Ikzf1 deficient zebrafish embryos rescued the defective lymphoid progenitor migration but not their posterior differentiation inside the thymus, suggesting that other factors were involved.

In medaka, two CCR9 paralogs, CCR9a and b, have been recently described. The first one is involved in the migration of progenitor cells into the thymus, whereas CCR9b is implicated in their positioning and progress in the subcapsulary area [78]. In this species, thymocytes can be classified as CCR9a^+^ cells and CCR9b^+^ cells. The first ones accumulate in the subcapsulary thymic area where they proliferate and begin to express CCR9b, directing the thymocyte migration towards the medullary area [76] where they interact with DCs [79] to presumably undergo negative selection. Therefore, CCR9 is assumed to fulfil the same role in the teleost thymus as several QKs do in mammals.

On the other hand, there is little information on the mechanisms and molecules involved in the egress of thymocytes from fish thymuses [79]. First, T cells leaving the zebrafish thymus seem to migrate to the kidney, whereas in medaka they appear in the intestine and the perivascular space of the trunk region [48]. Furthermore, in vivo imaging analysis suggests that thymic emigrants leave the medaka thymus through the same way as used to colonize the thymus [48].

In mice, there are four Notch genes, Notch 1–4, but only Notch 1 plays a clear role in the T cell differentiation with Notch 2 and 3, but not Notch 4, playing minor roles [73]. In teleosts, there are two orthologs of Notch 1, Notch 1a and Notch 1b, and one ortholog each for Notch 2 and Notch 3 [79]. Dll4 gene, the main ligand of Notch 1, is also duplicated in the genome of at least some teleosts [79].

Whole in situ hybridization (WISH) of the medaka thymus identifies Notch 1b in the outer zone where the CCR9a^+^ thymocytes accumulate after seeding the thymus. On the other hand, Notch 1b-/- larvae show reduced thymus size due to a decreased proliferation of lymphoid cells after thymus colonization. In addition, the Rag2 expression is reduced but neither the TcR rearrangements nor the selective processes undergone by thymocytes change [73]. These authors speculated that whereas Notch 1 is involved in all processes of murine T cell differentiation except the negative selection, Notch 1b plays a role in the first stages of lymphoid progenitor maturation including commitment and proliferation, and in the last ones during positive and negative selection, but not in the TcRβ recombination that governs the progress of DN thymocytes to DP cells. In this same way, DLL4 expressed in TECs would be necessary for the TcRβ expression [76]. Indeed, we know little on the intrathymic T cell selection in fish and, therefore, these results need further confirmation.

Teleosts appear to have a cytokine/cytokine receptor network quite similar to that of mammals [64,80], although there is no IL2Rα chain and both IL2 and IL15 must signal through IL15R [69,81]. However, a candidate for the IL7 gene, an important molecule for both T and B cell differentiation was identified in the zebrafish genome, but its role in fish lymphopoiesis appears to be limited [82]. Although IL7 is strongly expressed in the zebrafish thymus as well as in some areas of the renal lymphopoietic tissue, IL7 deficient fish are apparently normal, except for a reduced number of T precursor cells without changes in the proportions of αβ and γδ T cell lineages, in contrast to the situation in the murine thymus where the presence of γδ T cells is totally dependent on IL7 [83]. On the other hand, deficits in molecules belonging to the IL7 signaling pathway, such as IL7R and Jak3, cause a more severe phenotype. Remarkably, the IL7Rα chain forms a part of a second receptor, CRLF2, that in mammals is the receptor of another factor, TSLP, that has not been, however, identified in zebrafish. Thus, the absence of IL7Rα would affect two signaling pathways resulting in a more severe phenotype. In fact, CRLF2-like gene defective zebrafish show impaired T cell development that is less severe than that observed in IL7R-deficient mutants [84]. Together these results suggest that, apart from IL7, a proper T cell differentiation in zebrafish requires other unidentified cytokines. In this respect, Lawir and colleagues [82] concluded that the IL7 receptor could act in parallel with genes encoding IL2 and IL15 for governing zebrafish T cell differentiation, as morphants for IL2 plus IL15, or the IL15 receptor, but not those deficient in a single cytokine, resulted in a strong reduction in Rag1-expressing thymocytes.

Regarding T cell education, in jawed vertebrates the process is intimately related to the occurrence of MHC class I and class II molecules. In rainbow trout, the thymus is the first organ in which class I antigens are observed [85]. In *Dicentrarchus labrax*, the class II antigen expression increases in the thymus after lymphoid seeding and during the cortex/medulla compartmentalization. Class II β chain positive cells appear scattered throughout the cortex, presumably involved in positive selection but more frequently in the cortex/medulla limits and the medulla where the DCs are [38]. By using specific antibodies, class II antigens have been identified on TECs and hematopoietic cells (lymphocytes, macrophages and DCs) [86,87]. On the other hand, molecules linked to MHC function during thymic selection have also been reported in a few species of teleosts. Genes encoding homologous proteasomal β5t subunits occur in bony fish, together with those that encode the constitutive β5 and the inducible β5i form. Moreover, the protease Prss 16 exists in teleosts, as well as in Chondrycthies and lampreys [88], as well as tapasin and the transporters TAP1 and TAP2 [65,80,89]. In addition, Aire expression has been reported in the teleost thymic epithelium [90].

Allelic variation in classical MHC class I was conclusively related with allograft rejection [91,92]. In addition, bony fish exhibit MHC restriction: cytotoxic cells only kill virus-infected cells that share identical MHC class II molecules [93], but, unfortunately, to establish functional relationships between the existence in distinct thymic compartments of MHC molecules and the occurrence (or not) of thymocyte selection is difficult and the limited available evidence is indirect.

On the other hand, the nurse shark thymus exhibits activation-induced deaminases (AID) demonstrated by ISH and RT-PCR [94]. Their expression is co-localized with TcRα chain gene editing, a process that in mice has been related with the occurrence of thymic nurse cells [95], which we reported many years ago in trout thymuses [96].

Intimately related with T cell selection is the existence of T regulatory (Treg) cells in teleosts [97]. Zebrafish Foxp3a positive cells in the thymus are originated from αβ cells that express CD4. 1 and CD4. 2, but not CD8α, and in adult fish predominate in the kidney rather than in the thymus or spleen, and could mediate their immunosuppressive effects through IL10 secretion [97]. The zebrafish genome contains two Foxp3 orthologues, Foxp3a and Foxp3b [98,99,100], but the first one appears to be the functional equivalent of mammalian Foxp3 [97] because the proportions of zebrafish Foxp3a^+^ cells, but not those of Foxp3b^+^, increase when T cells develop [99,100]; in addition, transduced zebrafish Foxp3a in murine naïve CD4^+^ T cells results in immunosuppression after TcR stimulation [99], and in vivo Foxp3a deficient zebrafish exhibit, with age, infiltrations of mononuclear cells in peripheral tissues [98,100].

### 3.4. The Evolution of Foxn1/Foxn4 Family of Transcription Factors and the Thymic Epithelial Microenvironments

As previously mentioned, Foxn, a family of transcription factors, is critical for the maturation of thymic epithelial progenitor cells, providing molecules, such as IL7, chemokines and Dll4, and the Notch ligand, whose roles in the teleost thymopoiesis has been already reported. Nevertheless, there are some remarkable differences between the condition of the murine and lower vertebrate thymus: whereas in the first one, Foxn1 is the main regulatory factor, although it weakly expresses another member of the family, Foxn4, in teleosts, and even in primitive elasmobranchs (i.e., catshark, *Calorhinchus milii*), both Foxn1 and Foxn4 are expressed [101,102,103]. It has been proposed that an ancestral Foxn4 gene gave rise to the current Foxn4 gene of *Amphioxus* that, presumably just after vertebrate emergence, duplicated giving rise to the Foxn1 genes present in lower vertebrates [76]; later, Foxn4 was silenced, probably in the ancestors of tetrapods [103]. These authors proposed that the Foxn4 expression in the pharyngeal endoderm of the cephalochordate *Branchiostoma*, or perhaps of tunicates, predisposes them to further thymopoietic activity in this area [103]. 

Boehm and colleagues carried out an exhaustive study on the origins, evolution and capacities for supporting lymphopoiesis of the Foxn1/4 family by using an experimental model in which murine thymic epithelium developed under the control of Foxn1 and/or Foxn4 of different lower vertebrates [102,103,104]. When murine Foxn1-/- thymus developed under control of the Foxn4 gene of *Amphioxus*, lymphoid progenitor cells seeded the thymic primordium and differentiated until the DP thymocyte stage, but were unable to undergo TcR selection, in an epithelial microenvironment that resembled an immature cortical thymic epithelium able to produce Dll4 and CXCL12 [103,104]. The lack of SP thymocytes explains the absence of thymic medulla. When the murine thymus was governed by the *Callorhinchus* Foxn4 gene, a reduced number of T cells differentiated normally in a thymic microenvironment largely consisting of cortical epithelial cells, but the thymus contained a high number of immature B lymphocytes, principally in the mesenchyme of perivascular spaces, a finding also observed in thymus developed under murine Foxn4 control [102,104]. On the other hand, thymuses expressing *Callorhinchus* Foxn1 resemble those showing murine Foxn1, containing less immature B lymphocytes and a histologically distinguishable cortex and medulla. Remarkably, thymuses governed by both Foxn4 and Foxn1 maintained high numbers of B cells, suggesting that in any condition the presence of Foxn4 favors the generation of B lymphocytes [102,103].

The authors speculated that the ancestral vertebrate thymus would be a dual organ with both B and T lymphopoietic capacities that would occur in mesenchyme and epithelial environments, respectively. Apart from the scarce physiological models supporting this hypothesis, several questions remain unresolved. Why have current vertebrates not maintained this condition of T and B lymphopoiesis in the same organ? Remarkably, mature, but not developing, B lymphocytes have been found in the thymus of current cartilaginous [105] and bony fish [26,52,106], although current and extinct fishes would be different. According to the reported results, more than a mesenchyme or epithelial microenvironment, it is presumed that the Dll4 expression determines whether T or B cells should be differentiated. On the other hand, mesenchyme is a too broad non-specific term. For instance, two types of fibroblasts have been found in the murine thymus, capsulary and medullary, with different phenotypes and functions [107], and in the bone marrow that homes B lymphopoiesis in adult mammals, at least three mesenchyme-derived stromal cells (i.e., CAR cells, LeptinR cells and nestin cells) exist [27,108]. On the other hand, T cell specification via Dll4 has been pointed out as an evolutionary ancient function [76,102]. In this respect a comparative analysis of DLl4 expression in mice recovered with fish Foxn1 and/or Foxn4 would be illustrative. Finally, analysis of the behavior of the Foxn1 gene of lungfish and primitive tetrapods could provide interesting information.

A final issue that deserves interest is the relationship (if any) between the lamprey thymoid and the thymus of jawed vertebrates. Boehm and colleagues reported a lymphoid tissue at the tips of gill filaments in the gill basket of lamprey larvae, which was named the thymoid because of its possible analogies to the thymus of jawed vertebrates [109]. However, the thymoid is organized as anatomically separated, individual lymphoid organs, where variable lymphocyte receptors (VLR-A)^+^ and VLR-C^+^ cells infiltrate the gill epithelium constituting a lympho-epithelial organ, ultrastructurally different to the thymus. Remarkably, Takaba and colleagues failed to identify a similar organ in the other group of cyclostomes, the hagfish, which shares many immune characteristics with lampreys [110]. Key to correlating the lamprey thymoid and gnathostome thymus was that cytidine deaminases (CDA)-1 expressing VLR-A^+^ cells [109] and VLR-C^+^ cells [111] appear close to the Foxn^+^ gill epithelial cells. In addition, non-functional VLR-A gene assemblies occur only in the thymoids but not in other lymphoid tissues of lamprey (see below).

Undoubtedly, other characteristics of the lamprey thymoid are similar to the thymus of jawed vertebrates, such as the occurrence of a Foxn4^+^ epithelium in this gill area [109], the co-localization of DllB and Foxn4 expression [76], or the existence of CXCR4 [112] and CXCL12 genes [113] in the genome of several lamprey species, although CXCR4^+^ cells have not been found in the thymoid [76]. Other molecules, such as the CCL25/CCR9 pair, critical for attracting lymphoid progenitors into the thymus, have not been identified in lampreys.

Remarkably, although lampreys have neither MHC molecules nor know the mechanisms of antigen presentation in these primitive vertebrates, several results indirectly support that the VLR repertoire of lampreys would be selected in the thymoids of the gill area [114], because the repertoire in the thymoids and periphery are different: thymoids frequently contain non-functional VLR assemblies, but peripheral lymphoid cells not. In addition, VLR assemblies found in thymoids have different variable leucine-rich repeat (LRRV) modules, whereas in the periphery, these assemblies are fixed. Finally, as above mentioned, no functional VLR-C assemblies occur in the thymoids [115]. Obviously, the mechanisms of functioning of VLA and TcR are different and, consequently, it is not expected that thymoids and thymuses use the same processes to select their immune repertoires. Although precise mechanisms used for VLA selection in thymoids are unknown, it has been suggested that this occurs by negative selection, although this hypothesis needs confirmation [115]. On the other hand, it has been reported that VLR-A, a presumptive TcR equivalent, can recognize antigens directly [116].

In summary, we do not think that the jawed vertebrate thymus directly derives from the lamprey thymoid after the aggregation of several thymoids, but rather, a primitive non-vertebrate chordate containing a pharyngeal placode consisting of Foxn4^+^ epithelial progenitors evolved following two different, independent ways according to the anatomical organization of their pharyngeal regions, giving rise to a dispersed lymphoid tissue that occupies the tips of gill filaments in the complex pharynx of lampreys and to a single true thymus derived from one or several pharyngeal pouches after the emergence of jaws in the jawed vertebrates. This independent evolution provided two different microenvironments in thymoids and thymuses, allowing the differentiation of distinct types of immunocompetent cells.

### 3.5. The Dual Condition of Teleost Kidney as a Primary Lymphoid Organ and Active Participant in the Immune Responses Resemble That of the Bone Marrow of Higher Vertebrates

Peripheral (secondary) lymphoid organs are architecturally organized for providing an adequate spatial distribution of distinct types of immunocompetent cells for ensuring a proper immune responsiveness. In teleosts, the peripheral lymphoid organs are the kidney, the spleen and the mucosa-associated lymphoid tissues. Although some primitive lymphoid aggregates identified in some amphibians and reptiles were described as primitive lymph nodes, there is no evidence of their presence in ectothermic vertebrates and, really, they are well developed only in eutherian mammals [117].

The kidney is a major lympho-hematopoietic organ in teleosts. Anatomically, it constitutes a sheet occupying a dorsal position attached by connective tissue and mesothelium to the walls of the body cavity. The organ consists of a more anterior part named the pronephros or cephalic kidney, which contains blood-forming tissue and chromaffin tissue but not renal tubules, and a posterior or trunk kidney that, between the renal tubules, houses lympho-hematopoietic tissue. Therefore, both areas are histological and ultrastructurally similar [54,118,119] and show a mesenchyme reticular network that supports free hematopoietic cells. Some reticular cells, as well as endothelial cells lining blood sinusoids, exhibit phagocytic activity necessary for antigen processing and/or pathogen clearance [117]. 

Although some studies pointed out the teleost kidney as an equivalent of lymph nodes, which do not exist in lower vertebrates, its resemblance with mammalian bone marrow is assumed [120]. Accordingly, teleost kidneys show, as above mentioned, a dual function: in adult teleosts, renal lympho-hematopoietic tissue contains lymphoid progenitors [121] that seed the thymus, supports in situ B lymphopoiesis [122] and responds to antigens and/or pathogens.

Apart from developing and mature B cells, the teleost kidney contains T lymphocytes, DCs [87], macrophages, granulocytes, thrombocytes and melano-macrophages centers (MMCs) [117]. Both CD4 [123,124] and CD8 cells [125] have been described in both the spleen and kidney of several teleost fishes. In the kidney, CD8α cells significantly increase after the grafting of allogenic, but not isogenic, scales with a stronger response after second allografts [125]. Alloantigen-specific cytotoxic cells also increase after sensibilization with erythrocytes and skin grafting [126]. On the other hand, according to their profile of produced cytokines, the kidneys of teleost fishes contain TH1-like cells, TH2-like cells and TH17-like cells [64,127]. Furthermore, a higher number of Foxp3a positive T reg cells has been reported in the zebrafish kidney than in the spleen [97,99].

In teleost kidneys, lymphoid progenitors produce mature, naïve IgM^+^ cells and IgD^+^ cells that migrate to the mesonephros/trunk kidney and spleen [128]. It has been proposed that CCL20-L1a chemokine creates a gradient that favors the migration of naïve B cells from the pronephros to the mesonephros [30], but this hypothesis needs confirmation. Another point of discussion is the cycle of production, migration and final accumulation of long-term plasma cells (LT-PCs) in teleosts. In rainbow trout, B cells produce IgM^+^ antibodies 2 weeks after immunization with 2,4,6 trynitrophenyl-keyhole limpet hemocyanin (TNP-KLH) in pronephros but not in the spleen or peripheral blood, suggesting that this part of the kidney, as with mammalian bone marrow, houses LT-PCs [129,130,131,132]. However, other studies indicate that naïve B cells migrate to the spleen and posterior kidney after sensitization, differentiate to plasmablasts and PCs and then return to the pronephros [129,132]. On the other hand, in channel catfish, *Ictalurus punctatus*, immunized with TNP-KLH, Wu et al. [133] pointed out that most LT-PCs resided in or migrated to the anterior kidney, but a small fraction was located in the spleen and peripheral blood. Remarkably, it has been suggested that high affinity antibodies produced by LT-PCs could be selected in niches, whose nature is unknown, of the anterior kidney for maintaining the high affinity antibody titers [133], as occurs in bone marrow.

Although the spleen was considered the major thrombopoietic organ of teleosts because it contains both mature and developing thrombocytes, which increase after antigenic stimulation [134,135,136], currently it is assumed that the head and hind kidney are the main sources of thrombocytes in teleosts [137,138]. Remarkably, teleost thrombocytes take a part in immune responses [139,140]. They are involved, as in other non-mammalian vertebrates, in phagocytosis and intracellular killing of pathogens [139,140], express diverse components of the MHC, including MHC molecules of both class I and II, and act as antigen-presenting cells [136,139,140,141].

### 3.6. The Spleen, the Immunological Memory and the Absence of Germinal Centers in Teleosts

The vertebrate spleen is a blood filtering organ that, from an immunological view, evolves to adapt its anatomy, histology and function for a better management of the immune responses. In this respect, the delineation of a red and a white pulp, differentiation of T cell and B cell areas and presence of germinal centers (GCs) and marginal zones are important aspects for defining the evolution of splenic tissue. From its appearance in the most primitive jawed vertebrates, the splenic structure is importantly determined by the vascularization; thus, Tischendorff [142] emphasized that the lymphoid tissue of the spleen depends on the connection of the medium and small arteries with specific mesenchyme derivatives.

In teleosts, the splenic lymphoid tissue is poorly developed; in some species the spleen consists just of red pulp, presumably because the kidney is a major lymphoid organ in these bony fish that contains numerically more lymphocytes than the spleen [143] and splenectomy fails to avoid antibody responses against bovine serum albumin (BSA) in some teleosts [144]. Apart from both T and B lymphocytes, as mentioned in the previous section, the teleost spleen contains DCs, macrophages and a supporting reticular stroma. No distinguishable T and B cell areas have been identified in the spleen of numerous teleosts, but the total splenic lymphoid tissue increases after immunization. In addition, both ellipsoids and the named melano-macrophage centers (MMCs) are important structures for splenic immune responses. MMCs exist also in both the kidney and liver.

In all vertebrates, ellipsoids are terminal branches of splenic arteries that consist of reticular cells and macrophages within a framework of reticular cells that surrounds artery capillaries. They have been reported in most teleosts as splenic elements able to trap both antigenic and non-antigenic materials (Figure 3). MMCs are partially encapsulated aggregates of phagocytic cells associated with the vascular system, whose development varies in different organs and among distinct species. Thus, they are little developed in salmonids, appearing arranged throughout the lympho-hematopoietic tissue of the kidney and spleen and the periportal areas of the liver [145]. More or less developed MMCs exist also in amphibians and reptiles, particularly in the liver [117].

In all conditions, MMCs contain heterogeneous phagocytic materials, principally hemosiderin, melanin and lipofuscin [146,147], which, however, vary in relation to bleeding, vitamin deficiency, diet, starvation, senescence, disease, immunization, tissue homeostasis, etc. [117]. From an immunological view, splenic macrophages or phagocytic reticular cells in the kidney transport materials, including pathogens, to MMCs. As a consequence, antigens processed by macrophages accumulate in MMCs [147,148,149] (Figure 4). From these results some authors, but not others [122,150], hypothesized that teleost MMCs would be primitive, functional equivalents of mammalian germinal centers [151,152] involved in the generation of immunological memory, somatic hypermutation, B cell clone selection and increased antibody affinity.

Here we review the analogies and differences of the teleost MMCs with the mammalian GCs. The immunological memory is the ability of the immune system “to remember” antigens previously found and respond specifically to them stronger (quantitative and qualitatively) than after the first encounter [127]. This process is intimately related with the existence of long-term memory T and B lymphocytes, whose presence in teleostean lymphoid organs has been commented on in the previous section. Channel catfish and rainbow trout are two examples of teleosts showing immunological memory to T-dependent antigens (i.e., TNP-KLH), although presumably there are considerable differences among fish species [153]. Although the response is not equal to that in mammals, affinity maturation has conclusively been demonstrated in several teleosts (i.e., channel catfish, rainbow trout, Atlantic salmon, zebrafish) in response to T-dependent antigens [133]. Firstly, low affinity antibodies appear and later are substituted by high affinity antibodies [130,132,154]. The higher affinity antibodies remain longer and are maintained predominantly along the time [133], but the increase in affinity is lower (about 100 times) as compared to that observed in mammals [153].

In mammalian GCs, dividing B lymphocytes that undergo clonal selection after somatic hypermutation of their antigen receptor genes, with the concourse of follicular dendritic cells (FDC) and follicular helper T cells, defines the behavior of this B cell niche, whereas molecules of the tumor necrosis factor (TNF) superfamily (SF), particularly lymphotoxins, are necessary for the GC organization. Thus, a possible explanation for the lack of GCs in ectothermic vertebrates would be the absence of these molecules, but lymphotoxin genes have been identified in teleosts [155]. On the other hand, although the existence of follicular helper T cells in teleosts is uncertain, factors necessary for their functioning, such as IL6 and the transcription factor Bcl6 (B cell leukemia/lymphoma 6), have been reported in these fishes [64,156,157].

Studies supporting MMCs as primitive GCs are based on the occurrence of AID, B cells and CD4^+^ T cells in or close to these cell aggregates [151] and their capacity to store antigens, including those bound to antibodies [150,151]. It is true that MMCs trap and retain antigens to present to lymphocytes, but they have no capacity to organize a histologically identifiable GC. In addition, it is important to remark that many fishes including Chondrichthyes have small, isolated melano-macrophages that trap antigens but neither converge to create big MMCs nor exhibit antibody affinity maturation [158]. A similar condition presumably occurs in teleosts with small MMCs, such as salmonids, and in amphibians [117].

GCs are a consequence of the high frequency of mutated B cells, as reflects the high cell proliferation, the somatic hypermutation and increased antibody affinity that occur in these microenvironments. In lower vertebrates, including teleost fish, a lower B cell division capacity results in low numbers of high affinity antibodies and the absence of GC because low B cell expansion does not favor the activation of FDC precursors from lymphoid organ stroma and vice versa, making it impossible to organize the histological structure known as GCs. Thus, we reported in the adult trout spleen that repeated immunization with a rhabdovirus resulted in the appearance of small clusters of Rag+ cells, totally different from MMCs, which might represent areas for selecting few B cell clones, as small GCs [122].

In summary, MMCs are clearly clusters of macrophages involved in the homeostasis of fish tissues in normal and pathological conditions but not primitive GCs.

### 3.7. The Mucosal-Associated Lymphoid Tissues (MALT) in Teleost Fish

Mucosal immunity is particularly important in aquatic animals that live in environments filled up with antigens and pathogens. MALT (Mucosae-associated lymphoid tissue) in higher vertebrates consists of organs and well organized structures associated to mucosae that in teleosts are not so evident, and, therefore, we consider fish MALT to be a collection of diffuse lymphoid tissues [117]. The teleost MALT includes the SALT (skin), GALT (gut) and GIALT (gills) [159,160], including the interbranchial lymphoid tissue (ILT) [161,162] and, more recently, a NALT (nasopharynx) [163] and an organ located in the cloaca of the Atlantic salmon presumptively equivalent to the avian bursa of Fabricius [164].

Rather than providing a systemic description of the teleost MALT, we will finish this article devoted to the teleost lymphoid microenvironment, analyzing the most remarkable features of these organs that, according to the number of lymphocytes that they house, could be considered the main lymphoid organs of teleosts.

**SALT (Skin-associated lymphoid tissue)**. Apart from the presence of T and B lymphocytes [165,166], macrophages, granulocytes and mucus-producing secretory cells, the teleost epidermis contains Langerhans cells [167,168,169] and DC-like cells, which suggest a specific regulatory immune niche in the teleost skin. Langerhans cells in the epidermis of zebrafish contain Birbeck granules and express MHC class II antigens, TLR2 and langerin/CD207 [169]. Other authors have reported an MHC class II^+^CD8α^+^ cell subset in the total skin leucocyte population of several teleosts [170]. These cells correspond to semi-mature DCs and/or Langerhans cells, which express molecules involved in antigen processing (Lamp3, DC-SIGN) in MHC class I (Tapasin, calreticulin, ERp57) and class II contexts (Cathepsin), and co-stimulatory molecules, such as CD80/CD86, BAFF and CD40. Remarkably, these cells appear to have a dual immune capacity: they induce regulation of Foxp3 Treg cells in the absence of stimuli but provoke a cytotoxic T cell response after stimulation with TNP-KLH [170]. 

In teleost **GALT (Gut-associated lymphoid tissue)**, T and B lymphocytes, macrophages, neutrophils and eosinophils have been reported, as in other MALTs of teleosts. CD4-1^+^ T cells abound in the lamina propria of zebrafish gut and are scattered between the overlapping gut epithelial cells [97,171]. Remarkably, in the sea bass gut, the proportions of T lymphocytes are particularly high and, accordingly, some authors consider that it is the main lymphoid organ of adult fish [172]. In that species, there is a high heterogeneous expression of intestinal TcRα and TcRγ, low CD4 T cells and variable expression of both CD8α and MHC class II molecules [173]. These authors reported that sea bass intestinal intraepithelial lymphocytes (IELs) show in vitro spontaneous cytotoxicity and carry out in vivo Rag-controlling somatic rearrangements in the absence of antigen stimulation [174,175]. On the other hand, the so-called second segment of the gut has particular relevance in some teleosts. In this intestinal area, abundant amounts of antigens are trapped and transported from the lumen to intraepithelial macrophages [176], contributing to the induction of both mucosal and systemic immune responses, especially after anal immunization [177,178].

**Gills**. Remarkably, more TH2-like cells expressing the transcription factor Gata3 and IL4 than T-bet^+^IFNγ^+^ TH1-like cells have been found in zebrafish gills [171], whereas IgT remains the main Ig involved in the responses to pathogens in gills (see below) [165]. In teleost gills the process of antigen trapping and processing is a matter of discussion. Two types of cells have been reported in rainbow trout gills: macrophages/DCs, as above described in SALT, showing large vacuoles and expressing CD45, CD83, IL1β, IL12p40b and capacity for nanoparticle uptake [179], and UEA-1^+^ WGA- Anxa5^+^ cells resembling M cells of mammalian mucosae [180]. Remarkably, these fish M-like cells contain small vacuoles and express genes related to phagosomes/lysosomes and antigen processing. In addition, some of them express MHC class II molecules, suggesting that teleost gills could present antigens; on the contrary, M cells of both mammalian GALT and NALT process antigens via transcytosis. In other fish (i.e., zebrafish), although the gut epithelium traps nanoparticles and bacteria, there are not M-like cells [181].

The so-called ILT was described in 2008 in *Salmo salar* [161], as an intraepithelial accumulation of lymphocytes organized in two parts: proximal, at the end of the interbranchial septum, and distal in an extension of the trailing edge of the free filament, lining the whole branchial cleft of each gill [162]. Nevertheless, ILT shows important specie specific differences in distinct teleosts. Apparently, primitive teleosts possess ILT, while modern ones with reduced interbranchial septa have a lympho-epithelium in the base of the primary lamellae [179]. Histologically, the tissue is a lympho-epithelial, avascular organ containing principally T cells covered by a flat, non-keratinized squamous epithelium with some mucous cells and separated from the underlying tissue by a basement membrane [161,182]. ILT seems to be an epithelium infiltrated by lymphocytes rather than a primary lymphoid organ because it neither expresses Rag1 nor Aire, but contains Zap70^+^ T cells, CCL19^+^ cells, numerous MHC class II^+^ cells and IgT transcripts [179,182], and responds to infection exhibiting an increased size [183,184].

**NALT (Nasopharyngeal-associated lymphoid tissue)**. The structure and function of teleost NALT was recently reported by Sepahi and Salinas [185]. These authors, on the analysis of 4 families of teleosts, reported that NALT constitutes a non-organized, diffuse lymphoid tissue where IgT^+^ B cells predominate but also contains T cells, including Foxp3^+^ Treg cells [97]. Even in the absence of antigenic stimulation, the nasal secretion contains high values of IgT, which, after activation with diverse pathogens, increases significantly [163].

**Bursa of Fabricius equivalent**. Another locus of the MALT that was related with the avian bursa of Fabricius, has been recently reported in the cloacal region of the Atlantic salmon [164]. This “bursa” appears to be a diverticulum of the urogenital papilla that extends distally as a blind sac. It consists of a prominent epithelium continuous and histologically similar to the stratified squamous epithelium of the epidermis, which is notably infiltrated by lymphoid cells. In any case, despite its anatomical position, structurally this lymphoid area does not appear to be similar to the avian bursa. For instance, ISH analysis has demonstrated the presence of both T and B lymphocytes, and Rag-1 but not of AID or Rag-2. Some MHC class II^+^ cells appear within the bursal epithelium. Although the authors speculate that this fish bursal tissue could be a site for AID-induced somatic editing, we think that the location and the identified cell types suggest that this salmon bursa is a locus of antigen uptake and presentation, and the organ is therefore a key element for generating local immune responses [186] as with the avian caecal tonsils and their own bursa after birth [117].

The relevance of IgT/Z for the immune responses associated with mucosae in teleosts deserves a more extensive comment. Fish mucosal secretions contain the three classes of Ig reported to occur in teleosts, IgM, IgD and IgT/Z, but their contribution is different, and they are particularly enriched in IgT/Z [128,187,188]. 

Nevertheless, IgT/Z is lacking in 25 species out of 73 *Actinopterygii* [189], and in rainbow trout, three subclasses IgT1, IgT2 and IgT3 have been identified; all are present in serum [190]; IgT1 and IgT3 include membrane and secretory forms, but IgT2 is only secretory [24]. In addition, there are two subclasses, IgZ1 and IgZ2, in zebrafish. IgZ1 is present in both serum and the mucus of gills and skin, whereas IgZ2 only appears in the mucus [191]. On the other hand, IgT occurs as a monomer in the trout serum but as a polymer (4 or 5 monomers) joined by non-covalent interactions in the mucus [165,190,192].

Teleost GALT infected with different pathogens exhibits profiles of immune responses that are remarkably different. The number of IgT^+^ B cells as well as the IgZ titers significantly increased in the rainbow trout gut following infections with different pathogens, including parasites [190] and bacteria [193], while IgM^+^ B cells did not change. In other studies, both IgT and IgM transcripts increased in the rainbow trout midgut after *Yersinia ruckeri* administration [194]. Furthermore, infected rainbow trout with IPNV (infectious pancreatic necrosis virus), showed increased proportions of IgM^+^ and IgT^+^ B cells in the foregut and pyloric caeca but both in the hindgut [195]. More importantly, IgT-depleted fish are more susceptible to mucosal pathogens because IgM and/or IgD exert no compensatory roles [196]. On the other hand, in several mucosae, IgT is the major antibody for coating the microbiota and, in its absence, teleost fish suffer profound dysbiosis [188].

A similar behavior is shown in the lymphoid tissue of the gills and nasopharynx. In general, there is an increase in the IgT^+^ B cells in response to several parasites [165], bacteria [191,197] and viruses [198], whereas the IgM^+^ B cell numbers remain unchanged. The opposite condition occurs in the serum. In the skin, diverse parasites induce specific IgT but irrelevant IgM responses. Mucus IgT covers the skin and IgT^+^ B cells, but not IgM^+^ B cells, accumulate in the epidermis [199,200]. Only in the Atlantic salmon infected with *Lepeophtheirus salmonis* did Tadiso and colleagues [201] report higher upregulated expression of IgM than IgT. Likewise, diverse bacteria induced increased IgT transcripts in the skin with little or no IgM-mediated response [202,203].

## 4. Conclusions

It is evident that important advances have taken place in teleost immunology during the last years. They emphasize, among other topics, the relevance of the structure of lymphoid organs where immunocompetent cells live for their immunological jobs. However, some questions remain unanswered and need further research. For example, there are inconsistencies about the origin of the thymus and its relationship with the changes undergone by the pharyngeal region in jawless and jawed fish. Likewise, why the teleost thymus remains in continuity with the pharyngeal epithelium in adult teleosts is an unresolved issue. It is important also to review the role played by TECs in the processes of the positive and negative selection of thymocytes. Presumably, they occur in teleosts but our knowledge of the cells, molecules and underlying mechanisms is still limited.

Undoubtedly, the absence of lymph nodes in ectotherms is due to the little or no development of their lymphatic system; in this respect, the kidney and other organs in ectotherms [204] illustrate the relevance of cell microenvironments for governing the behavior of the lympho-hematopoietic tissue. As shown by the reported results, independently of the organ and other functions, any organ can substitute the bone marrow as a blood-forming organ when it contains a similar stroma.

A better knowledge of the mechanisms of antigen trapping and processing in the teleost spleen as well as the involved cell subsets is necessary. The absence of germinal centers must be considered as just the reflection of a low efficiency in the selection of B cell clones rather than due to the lack of follicular dendritic cells (FCD) or follicular helper T cells in teleosts.

The analysis of antigen trapping in MALT, conclusively identifying the involved cells, will improve the knowledge on fish mucosal immunity and the efficiency of vaccines. However, presumably, many of these proposals may be accepted only when more specific reagents, particularly monoclonal antibodies, are available. In this respect, we have added a couple of Tables to summarize the principal cells (Appendix A) and molecules (Appendix A) in teleost immunity, but it is important to remember that they are related just to a few species, the most extensively studied.

## Figures and Tables

**Figure 1 biology-11-00747-f001:**
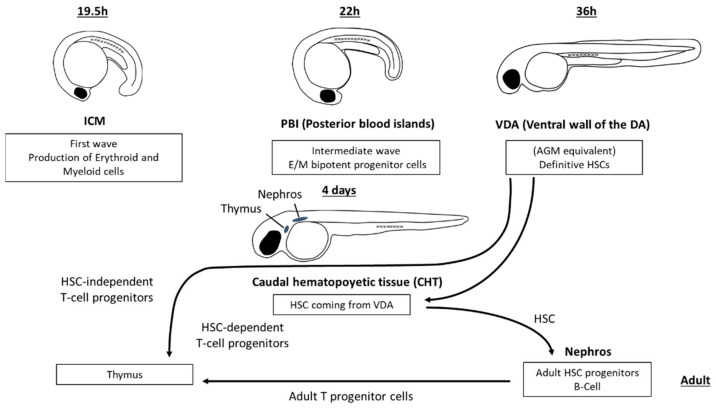
Lympho-hematopoietic sites in the developing zebrafish. First haemopoietic loci include ICM (Intercellular mass), PBI (posterior blood islands) and VDA (ventral wall of the dorsal aorta). From this last region, definitive HSC (hematopoietic stem cells) colonize the caudal hematopoietic tissue (CHT), and both HSC-independent and HSC-dependent T cell progenitors seed into the thymic primordium. From CHT, progenitor cells colonize the nephros that is the main hematopoietic organ of adult teleosts.

**Figure 2 biology-11-00747-f002:**
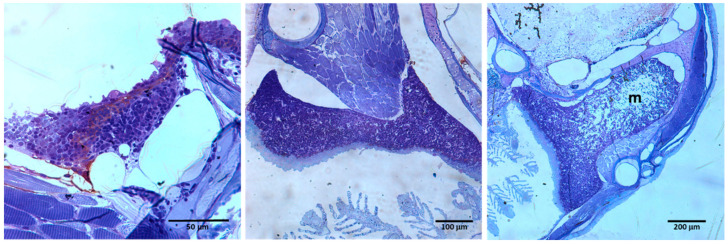
Serial sections of 3-month-old zebrafish thymus. Note that it is only possible to distinguish a central medulla (m) surrounded by a cortex densely occupied by thymocytes when the section shape changes.

**Figure 3 biology-11-00747-f003:**
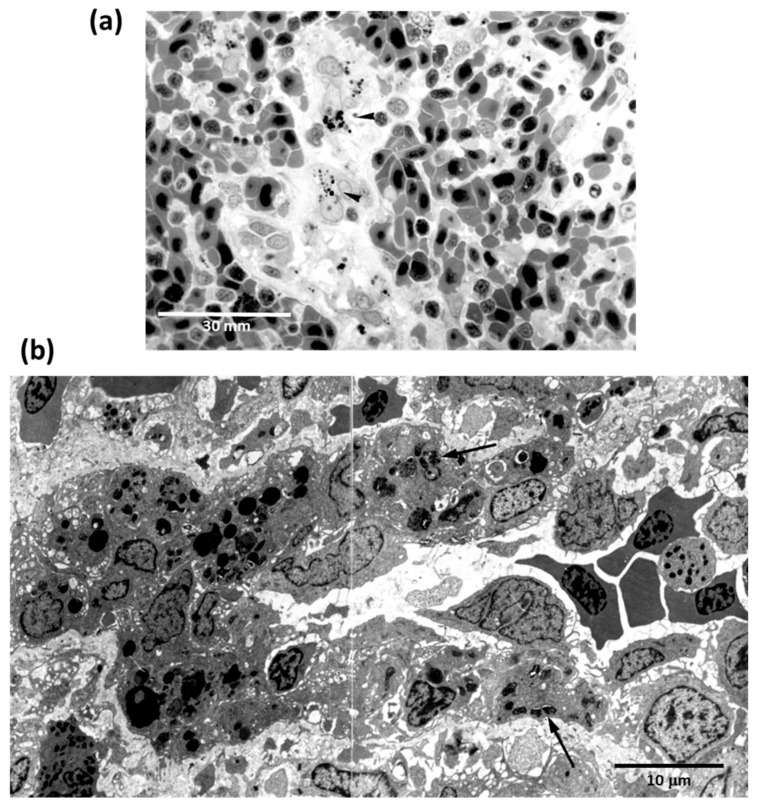
Light microscopy of the carbon uptake (arrowheads) by macrophages surrounding the splenic ellipsoids of *Carassius auratus* (**a**). By electron microscopy, ellipsoid macrophages (arrows) appear full off engulfed bacteria after intraperitoneal injection of *Yersinia ruckeri* (**b**).

**Figure 4 biology-11-00747-f004:**
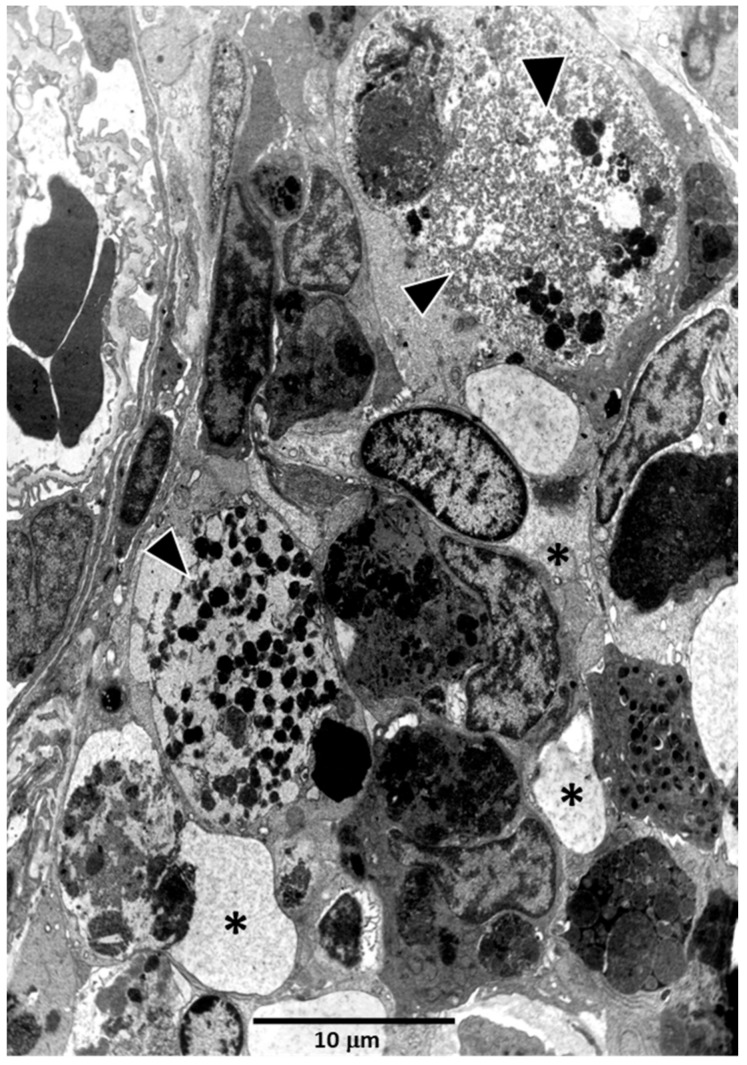
A melano-macrophage center close to the renal glomerulus of *Carassius auratus.* Note the presence of isolated melano-macrophages (arrowheads) and other phagocytic cells (asterisks) full of cell debris.

## Data Availability

Not applicable.

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
