# Peer review of "Lympho-Hematopoietic Microenvironments and Fish Immune System"

_biology, 2022, doi:10.3390/biology11050747_

Round 1

Reviewer 1 Report

The review of Lympho‐hematopoietic microenvironments and fish immune system written by Agustin was very nice decryption with reference to the evolutionary approach as well as mentioning most lymph-related organs. Just adding a couple of things, even though it was accepted as important, the thymus part felt wordy due to the detailed and long explanation. So hopefully it would be recommended to make a concise explanation. Another thing is that it may raise the understanding of this review by adding a couple of diagrams that will help readers get images and appreciation. Lastly, it is a little bit felt that there are fewer data based on cell markers such as specific monoclonal antibodies.

Author Response

  • As suggested, I have reduced the sections 3.1 to 3.5)(see changes in red from page 4 to page 10) on teleost thymus trying to remark the main topics and avoiding others, for a more concise explanation and, therefore, more comprehensible
  • It is difficult to do a Table including teleost immune cells based on the marker expression detected by specific reagents because data are few, the detection is frequently based on cross-reactivity and specially are concerned with a few species making difficult to reach general conclusions. Nevertheless, in agreement with the referee’s suggestions, we have added two Tables (S1, S2) as supplementary material at the end of the manuscript (changes are marked on red), summarizing the immune cell types described in the text, some of their characteristics and the main molecules that govern their development and function
  • As suggested, I have reduced the sections 3.1 to 3.5)(see changes in red from page 4 to page 10) on teleost thymus trying to remark the main topics and avoiding others, for a more concise explanation and, therefore, more comprehensible
  • It is difficult to do a Table including teleost immune cells based on the marker expression detected by specific reagents because data are few, the detection is frequently based on cross-reactivity and specially are concerned with a few species making difficult to reach general conclusions. Nevertheless, in agreement with the referee’s suggestions, we have added two Tables (S1, S2) as supplementary material at the end of the manuscript (changes are marked on red), summarizing the immune cell types described in the text, some of their characteristics and the main molecules that govern their development and function

Reviewer 2 Report

Congratulations Dr. Zapata for the excellent paper, this review discusses important points in the immune system ontogenesis of teleost fish, mainly addressing their lymphoid organs, as well as establishing phylogenetic comparisons with other vertebrates, becoming an important tool in the identification of key points in the adaptive immunity of these animals.

Author Response

No comments

Reviewer 3 Report

The present manuscript is a comprehensive review of the origin and development of the fish immune systeμ. The work covers all the issues related to the evolution of the immune system of fish and compares the data so far with the already existing knowledge about the immune system of mammals. I have only brief remarks, which are intended to help the reader understand the complex concepts described in the text. Specifically, author should add:

  1. A figure describing histological organization showing cell populations in paragraph 3.1.
  2. A table showing the T and B cell populations (surface molecules, receptors, cytokines and chemokines produced, action).
  3.  Etc... should be removed.
  4. All abbreviations should be written in full the first time mentioned.

Author Response

1) Honestly, I don´t understand the referee requirement. Histological organization of the teleost thymus is described in the text of the original manuscript, emphasizing those aspects more specific of the teleost thymus, such as its continuity with the gill epithelium and the difficulties to distinguish thymic cortex and medulla. In my opinion both issues are extensively discussed in the original manuscript, and the relevance of the special shape of the organ for distinguish thymic cortex and medulla illustrated in the figure 2.

On the other hand, it is difficult to perform a general histological interpretation of the teleost thymus because as described, information on the types of thymic epithelial cells (TECs) and their arrangement in the different thymic areas is controversial when different teleosts are compared and even when the same species is analysed. For example, I described by electron microscopy only one TEC type in the thymus of Rutilus rutilus (see reference 50), but seven by immunohistochemistry in the thymus of rainbow trout (see reference 46). Also, I could include several by electron microscopy of zebrafish thymus but they will be similar to other ones published many years ago and cited in the current review (see references 6, 36, 50). In summary, a recommend to see the diagram on the rainbow trout thymus included in the reference 46.

2) We are included two Tables as supplementary material at the end of manuscript summarizing the most important immunocompetent cells described in teleosts as well as the molecules that govern their development and function. Nevertheless, it is difficult to reach general conclusions from these data because they are restricted to a few group of teleosts

3) The “Etc” term has been removed from the text

4) We have checked again the text and now all abbreviations are written in full the first time mentioned (introduced changes are marked in red)